# Learning Disentangled Representations in Deep Generative Models

**N. Siddharth, Brooks Paige, Alban Desmaison, Frank Wood & Philip Torr**
Department of Engineering Science, University of Oxford, Oxford OX13PJ, UK
`{nsid,brooks,alban,fwood,phst}@robots.ox.ac.uk`

**Jan-Willem van de Meent**
College of Computer Science
Northeastern University
MA 02115, USA
`j.vandemeent@northeastern.edu`

**Pushmeet Kohli**
Microsoft Research
WA 98052, USA
`pkohli@microsoft.com`

**Noah D. Goodman**
Department of Psychology
Stanford University
CA 94305, USA
`ngoodman@stanford.edu`

## ABSTRACT

Deep generative models provide a powerful and flexible means to learn complex distributions over data by incorporating neural networks into latent-variable models. Variational approaches to training such models introduce a probabilistic encoder that casts data, typically unsupervised, into an entangled representation space. While unsupervised learning is often desirable, sometimes even necessary, when we lack prior knowledge about what to represent, being able to incorporate domain knowledge in characterising certain aspects of variation in the data can often help learn better disentangled representations. Here, we introduce a new formulation of semi-supervised learning in variational autoencoders that allows precisely this. It permits flexible specification of probabilistic encoders as directed graphical models via a stochastic computation graph, containing both continuous and discrete latent variables, with conditional distributions parametrised by neural networks. We demonstrate how the provision of dependency structures, along with a few labelled examples indicating plausible values for some components of the latent space, can help quickly learn disentangled representations. We then evaluate its ability to do so, both qualitatively by exploring its generative capacity, and quantitatively by using the disentangled representation to perform classification, on a variety of models and datasets.

## 1 INTRODUCTION

Reasoning in complex perceptual domains such as vision often requires the ability to effectively learn flexible representations of high-dimensional data, interpret the representations in some form, and understand how the representations can be used to reconstruct the data. The ability to learn representations is a measure of how well one can capture relevant information in the data. Being able to interpret the learned representations is a measure of extracting consistent meaning in an effort to make sense of them. Having the ability to reliably reconstruct the data, a tool for predictive synthesis, can aid in model diagnosis, enable successful transfer learning, and improve generality. Such tasks are typically best addressed by generative models, as they exhibit the flexibility required to satisfy all three facets. Discriminative models primarily attend to the first two, learning flexible representations and conforming to some interpretable space (e.g. classification domain) but don't perform the predictive synthesis task.

Probabilistic graphical models (Koller & Friedman, 2009; Murphy, 2012) are a framework for generative modelling that enables specifying a joint probability distribution on a richly semantic representation space. As good a fit as they are for specification and representation, the learning process for both the analysis and synthesis tasks typically suffers in complex perceptual domains such as vision. This is because constructing a generative model requires explicitly specifying the conditional distribution of the observed data given latent variables of interest. In practice, designing such

likelihood functions by hand is incredibly challenging, and applying generative models to vision data often requires extensive and significant feature engineering to be successful. One approach to alleviate some of this hardship involves the development of deep generative models: generative models that employ neural networks to learn, automatically from data, the unknown conditional distribution in the model. They function as flexible feature learners, where the features are encoded in the posterior distribution over the latent variables in the model. Recent work exploring the effectiveness of such models (e.g. Kingma & Welling (2014); Kulkarni et al. (2015b); Goodfellow et al. (2014)) has shown considerable promise in being able to address the fundamental issues in performing this task. These models however are typically unsupervised, learning representations that are not directly amenable to human interpretation. Any interpretability or disentanglement of the learned representation is observed or extracted *after* learning has been performed, by exploring the latent space along its non-specific axes of variation. A more recent approach by Chen et al. (2016) involves imposition of information-theoretic constraints to better separate factors of variation, but here too, any interpretability is only established post facto.

While such approaches have considerable merit, particularly when faced with the absence of any information about the data, when there are aspects of variation in the data that *can* be characterised effectively, using and being able to express these can often be desirable. For example, when learning representations for images of house numbers, having an explicit "digit" latent variable helps capture a meaningful axis of variation, independent of other aspects. We also often want to interpret the same data in different ways depending on context: for a given image of a person, do we care about the identity, lighting, or indeed any other facets of the scene (c.f. Figure 1). In these situations, not being able to enforce context is something of a handicap.

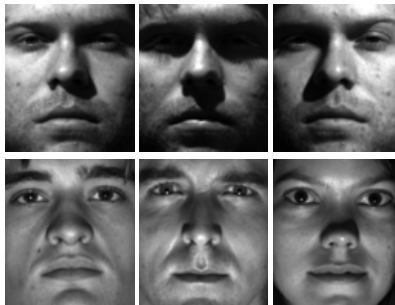

Figure 1: Variation along (top) lighting and (bottom) identity axes.

In this paper, we seek to combine the best of both worlds: providing the facility to describe the structural constraints under which we would like to interpret the data, while using neural nets to capture variation for aspects we cannot, or choose not to, explicitly model. By structural constraints, we refer to the (arbitrary) dependencies one would like to employ in the recognition model, particularly in regard to there being consistent interpretable semantics of what the variables in the model represent. In particular, we set up our framework in the context of variational autoencoders (VAE Kingma & Welling (2014); Rezende et al. (2014)), as a means for semi-supervised learning in deep generative models (Kingma et al., 2014). We provide an alternate formulation of the variational objective and a modified training procedure which permits us to explore a wide space of recognition networks to use as probabilistic encoders. In particular we make no mean-field assumptions for our recognition networks, allowing arbitrary hierarchical and structured-graphical-model representations, employing both continuous and discrete latent variables that can be alternately observed, or left unobserved.

## 2 BACKGROUND AND RELATED WORK

Variational autoencoders (Kingma & Welling, 2014; Rezende et al., 2014) simultaneously train both a probabilistic encoder and decoder for a dataset $\mathbf{x}$. The central idea is that an encoding $\mathbf{z}$ can be considered a latent variable which allows describing a decoder as a conditional probability density $p_\theta(\mathbf{x}|\mathbf{z})$. This is typically a distribution with parameters defined as the output of a deterministic multi-layer neural network (itself with parameters $\theta$) which takes $\mathbf{z}$ as input. Placing a weak prior over $\mathbf{z}$, the corresponding probabilistic encoder can be interpreted as the posterior distribution $p_\theta(\mathbf{z} \mid \mathbf{x}) \propto p_\theta(\mathbf{x} \mid \mathbf{z})p(\mathbf{z})$. Estimating parameters $\theta$ in this model is challenging, as is performing the posterior inference necessary to encode data. The variational Bayes approach learns an approximate encoder $q_\phi(\mathbf{z} \mid \mathbf{x})$, called an "inference network" or a "recognition network", which aims to approximate the posterior distribution $p_\theta(\mathbf{z} \mid \mathbf{x})$. Then, rather than fitting parameters $\theta$ by maximizing the marginal likelihood $p_\theta(\mathbf{x})$, the variational approach maximizes an *evidence lower bound* (ELBO) $\mathcal{L}(\phi, \theta; \mathbf{x}) \leq \log p_\theta(\mathbf{x})$, defined with respect to both decoder $\theta$ and encoder $\phi$ parameters.

$$\mathcal{L}(\phi, \theta; \mathbf{x}) = \mathbb{E}_{q_\phi(\mathbf{z}|\mathbf{x})}[\log p_\theta(\mathbf{x}, \mathbf{z}) - \log q_\phi(\mathbf{z} \mid \mathbf{x})], \tag{1}$$

One line of work to embed structure into the latent space $\mathbf{z}$ such that it exhibits disentangled features, is through partial supervision. This is either in terms of labelled data (Sohn et al., 2015),

or curriculum-learning schemes (Kulkarni et al., 2015b) which explicitly disentangle different factors. Kingma et al. (2014) explore semi-supervised learning in the VAE setting by factoring the latent space to learn a joint classification model $q_\phi(\mathbf{y} \mid \mathbf{x})$ and recognition model $q_\phi(\mathbf{z} \mid \mathbf{x})$. This is done by separating the latent space into structured, interpretable components $\mathbf{y}$ and unstructured components $\mathbf{z}$, analytically marginalising variables out where discrete. Sohn et al. (2015) perform fully-supervised learning in VAEs by transforming an unconditional objective into one where the data conditions both the (unstructured) latent and the (structured) labels. In contrast to Kingma et al. (2014), the learning objective is a lower bound on the conditional marginal likelihood $p_\theta(\mathbf{x} \mid \mathbf{y})$, conditioning the learned VAE on the values of the labelled data. Both of these approaches effectively require the label space $\mathbf{y}$ to be discrete and finite. Kulkarni et al. (2015b) attend to weakly-supervised learning with VAEs through a novel training procedure that uses data clustered into equivalence classes along different axes of variation. They then constrain different parts of the latent space to account for changes along a *single* axis, by training with data from a particular equivalence class. An advantage of this approach is not requiring any explicit labels on the latent space, though it does require independence assumptions on structured components, as well as carefully curated data.

An alternative approach biases towards interpretable representations by introducing structure in the prior distribution over the latent space $p(\mathbf{z})$. Johnson et al. (2016) explore the combination of graphical models and VAEs using classical conjugate exponential family statistical models as structured priors over the latent space. They consider relaxation of conjugacy constraints in the likelihood model using neural network approximations, with a training scheme resembling traditional mean-field coordinate ascent algorithms. The recognition network, rather than proposing values outright, proposes parameters of a conjugate-likelihood approximation to the true non-conjugate likelihood.

From a specific-instance perspective, Eslami et al. (2016) use a *recurrent neural network* (RNN) coupled with a *spatial transformer network* (STN, Jaderberg et al. (2015)) inducing a particular state-space representation with the approximation distribution of a VAE to parse images into scene constituents. Kulkarni et al. (2015a) also explore a specific instance related to a 3D graphics engine by having a programmatic description provide structure using neural networks as surrogates for the perceptual-matching problem. Andreas et al. (2016) explore a more general formulation of structure with compositional neural network models derived from linguistic dependency parses.

## 3 FRAMEWORK AND FORMULATION

Our method synthesises the semi-supervised and structured-graphical-model approaches. Like Johnson et al. (2016), we incorporate graphical model structures, however rather than placing them within the generative model $p_\theta(\mathbf{z}, \mathbf{x})$, we incorporate them into the *encoder model* $q_\phi(\mathbf{z} \mid \mathbf{x})$. For many perceptual problems in domains such as vision, complex dependencies arise in the posterior due to deterministic interactions during rendering. A mean-field approximation in $q_\phi(\mathbf{z} \mid \mathbf{x})$ is a poor fit, even in situations where all the interpretable latent variables are *a priori* independent. This is an important reason for our choice of where we embed structure. The use of a structured, multilevel probabilistic model to define the encoder can also be interpreted as a hierarchical variational model (Ranganath et al., 2015). Interpretability is enforced by occasionally supplying labels to latent variables expected to have a interpretable meaning in the final encoded representation.

Our framework provides an embedded domain-specific language (EDSL) in Torch (Collobert et al., 2011), that can be used to specify a wide variety of graphical models in the form of a stochastic computation graph (Schulman et al., 2015). An example is shown in Figure 2. These graphical models describe the structure of latent, observable, and partially observable random variables which exist in an idealized representation space. Specifically, we assume a model structure of the form $p_\theta(\mathbf{x}, \mathbf{z}, \mathbf{y}) = p_\theta(\mathbf{x} \mid \mathbf{z}, \mathbf{y})p(\mathbf{z}, \mathbf{y})$ where the likelihood $p_\theta(\mathbf{x} \mid \mathbf{z}, \mathbf{y})$ of the data $\mathbf{x}$ is conditioned on a set of *structured* variables $\mathbf{y}$ and *unstructured* variables $\mathbf{z}$, for which we define some appropri-

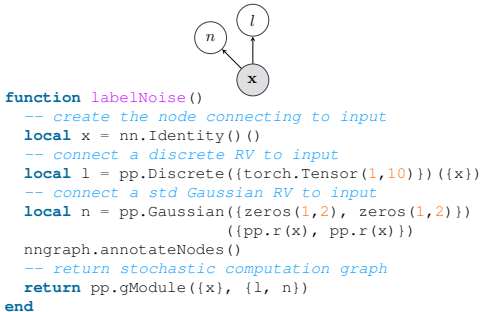

```
function labelNoise()
  -- create the node connecting to input
  local x = nn.Identity()()
  -- connect a discrete RV to input
  local l = pp.Discrete({torch.Tensor(1,10)})({x})
  -- connect a std Gaussian RV to input
  local n = pp.Gaussian({zeros(1,2), zeros(1,2)})
                         ({pp.r(x), pp.r(x)})
  nngraph.annotateNodes()
  -- return stochastic computation graph
  return pp.gModule({x}, {l, n})
end
```

Figure 2: Example graphical model and its expression in our framework. Further details in the Appendix.

ately structured prior $p(\mathbf{z}, \mathbf{y})$. The likelihood itself is typically unstructured (e.g. a multivariate normal distribution). This model structure allows us to optimize the parameters $\theta$ learning a likelihood function constrained by the structured latents, but crucially does not require that these latents completely explain the data. The approximation to the true posterior is nominally taken to be of the form of the prior distribution $q_\phi(\mathbf{z}, \mathbf{y} \mid \mathbf{x})$, with parameters $\phi$ but can often include additional structure and alternate factorisations as appropriate. Models with such factoring are useful for situations where interpretability is required, or informative, for some axes of variation in the data. It is also useful when we wish to interpret the same data from different contexts and when we cannot conceivable capture all the variation in the data due to its complexity, settling for particular restrictions, as is often the case with real world data.

A particular challenge here lies in choosing a manner for incorporating labelled data for some of the $\mathbf{y}$ into a training scheme. For example, choosing $q_\phi(\mathbf{z}, \mathbf{y} \mid \mathbf{x}) = q_{\phi_\mathbf{z}}(\mathbf{z} \mid \mathbf{y}, \mathbf{x}) q_{\phi_\mathbf{y}}(\mathbf{y} \mid \mathbf{x})$, decomposes the problem into simultaneously learning a classifier $q_{\phi_\mathbf{y}}(\mathbf{y} \mid \mathbf{x})$ alongside the generative model parameters $\theta$ and encoder $q_{\phi_\mathbf{z}}(\mathbf{z} \mid \mathbf{x}, \mathbf{y})$. In the fully unsupervised setting, the contribution of a particular data point $\mathbf{x}^i$ to the ELBO can be expressed, with minor adjustments of Equation (1), as

$$\mathcal{L}(\theta, \phi; \mathbf{x}^i) = \mathbb{E}_{q_\phi(\mathbf{z}, \mathbf{y} \mid \mathbf{x}^i)} \left[ \log \frac{p_\theta(\mathbf{x}^i \mid \mathbf{z}, \mathbf{y}) p(\mathbf{z}, \mathbf{y})}{q_{\phi_\mathbf{z}}(\mathbf{z}, \mathbf{y} \mid \mathbf{x}^i)} \right]. \tag{2}$$

a Monte Carlo approximation of which samples $\mathbf{y} \sim q_{\phi_\mathbf{y}}(\mathbf{y} \mid \mathbf{x})$ and $\mathbf{z} \sim q_{\phi_\mathbf{z}}(\mathbf{z} \mid \mathbf{y}, \mathbf{x})$.

By contrast, in the fully supervised setting the values $\mathbf{y}$ are treated as *observed* and become fixed inputs into the computation graph, instead of being sampled from $q_\phi$. When the label $\mathbf{y}$ is observed along with the data, for fixed $(\mathbf{x}^i, \mathbf{y}^i)$ pairs, the lower bound on the *conditional* log-marginal likelihood $\log p_\theta(\mathbf{x} \mid \mathbf{y})$ is

$$\mathcal{L}_{\mathbf{x} \mid \mathbf{y}}(\theta, \phi_z; \mathbf{x}^i, \mathbf{y}^i) = \mathbb{E}_{q_{\phi_z}(\mathbf{z} \mid \mathbf{x}^i, \mathbf{y}^i)} \left[ \log \frac{p_\theta(\mathbf{x}^i \mid \mathbf{z}, \mathbf{y}^i) p(\mathbf{z} \mid \mathbf{y}^i)}{q_{\phi_z}(\mathbf{z} \mid \mathbf{x}^i, \mathbf{y}^i)} \right]. \tag{3}$$

This quantity can be optimized directly to learn model parameters $\theta$ and $\phi_\mathbf{z}$ simultaneously via SGD. However, it does not contain the encoder parameters $\phi_\mathbf{y}$. This difficulty was also encountered in a related context by Kingma et al. (2014). Their solution was to augment the loss function by including an explicit additional term for learning a classifier directly on the supervised points.

An alternative approach involves extending the model using an auxiliary variable $\tilde{\mathbf{y}}$. Defining $p(\tilde{\mathbf{y}}, \mathbf{y}, \mathbf{z} \mid \mathbf{x}) = p(\tilde{\mathbf{y}} \mid \mathbf{y}) p(\mathbf{x}, \mathbf{y}, \mathbf{z})$ and $q(\tilde{\mathbf{y}}, \mathbf{y}, \mathbf{z} \mid \mathbf{x}) = p(\tilde{\mathbf{y}} \mid \mathbf{y}) q(\mathbf{y}, \mathbf{z} \mid \mathbf{x})$, with likelihood $p(\tilde{\mathbf{y}} \mid \mathbf{y}) = \delta_{\tilde{\mathbf{y}}}(\mathbf{y})$, we obtain a model for which marginalization over $\tilde{\mathbf{y}}$ reproduces the ELBO in Equation (2), and treating $\tilde{\mathbf{y}}$ as observed gives the supervised objective

$$\mathcal{L}(\theta, \phi; \mathbf{x}^i) \big|_{\tilde{\mathbf{y}} = \mathbf{y}^i} = \mathbb{E}_{q_{\phi_\mathbf{y}}} \left[ \delta_{\mathbf{y}^i}(\mathbf{y}) \, \mathbb{E}_{q_{\phi_\mathbf{z}}} \left[ \log \frac{p_\theta(\mathbf{x}^i \mid \mathbf{z}, \mathbf{y}) p(\mathbf{z}, \mathbf{y})}{q_{\phi_\mathbf{y}}(\mathbf{y} \mid \mathbf{x}^i) q_{\phi_\mathbf{z}}(\mathbf{z} \mid \mathbf{y}, \mathbf{x}^i)} \right] \right]$$

$$= q_{\phi_\mathbf{y}}(\mathbf{y}^i \mid \mathbf{x}^i) \, \mathbb{E}_{q_{\phi_\mathbf{z}}} \left[ \log \frac{p_\theta(\mathbf{x}^i \mid \mathbf{z}, \mathbf{y}^i) p(\mathbf{z}, \mathbf{y}^i)}{q_{\phi_\mathbf{y}}(\mathbf{y}^i \mid \mathbf{x}^i) q_{\phi_\mathbf{z}}(\mathbf{z} \mid \mathbf{y}^i, \mathbf{x}^i)} \right]$$

$$= q_{\phi_\mathbf{y}}(\mathbf{y}^i \mid \mathbf{x}^i) \left[ \mathcal{L}_{\mathbf{x} \mid \mathbf{y}}(\theta, \phi_\mathbf{z}; \mathbf{x}^i, \mathbf{y}^i) + \log p(\mathbf{y}^i) - \log q_{\phi_\mathbf{y}}(\mathbf{y}^i \mid \mathbf{x}^i) \right]. \tag{4}$$

This formulation enables a range of capabilities for semi-supervised learning in deep generative models. To begin with, it extends the ability to partially-supervise latent variables to those that have continuous support. This effectively learns a regressor instead of a classifier in the same formulation. Next, it automatically balances the trade-off between learning a classifier/regressor and learning the parameters of the generative model and the remainder of the recognition network. This is due to the fact that the classifier $q_{\phi_\mathbf{y}}(\mathbf{y} \mid \mathbf{x})$ is always present and learned, and is contrast to the hyperparameter-driven approach in Kingma et al. (2014). Finally, it allows for easy automatic implementation of a wide variety of models, separating out the labelled and unlabelled variables, to derive a unified objective over both the supervised and unsupervised cases. When unsupervised, the value of the label $\mathbf{y}^i$ is sampled from $q_{\phi_\mathbf{y}}(\mathbf{y} \mid \mathbf{x})$ and scored in that distribution, and when supervised, it is set to the given value, and scored in the same distribution. This is in the same spirit as a

number of approaches such as *Automatic Differentiation* (AD) and Probabilistic Program inference, where the choice of representation enables ease of automation for a great variety of different cases.

**Supervision rate.** While learning with this objective, we observe data in batches that are either wholly supervised, or wholly unsupervised. This typically obviates the need to construct complicated estimators for the partially observed cases, while also helping reduce variance in general over the learning and gradient computation (details of which are provided in the Appendix). Doing so also presents a choice relating to *how often* we observe labelled data in a complete sweep through the dataset, referred to as the *supervision rate* $r$. Practically, the rate represents a clear trade-off in learning the generative and recognition-network parameters under interpretability constraints. If the rate is too low, the supervision can be insufficient to help with disentangling representation in the recognition network, and if too high, the generative model can overfit to just the (few) supervised data points. The rate also has a natural relation to the variance of the objective function and its gradients. As can be seen from Equation (4), an evaluation of the objective for a given $\mathbf{y}^i$ involves the unsupervised estimation of the conditional ELBO $\mathcal{L}_{\mathbf{x}|\mathbf{y}}$. The rate implicitly affects the number of such estimations for any given $\mathbf{y}^i$ and thus the variance of the objective with respect to that label $\mathbf{y}^i$. The same argument applies for the gradients of the objective.

**Plug-in estimation for discrete variables.** In targeting a general class of models, another particular difficulty is the ubiquity of discrete latent variables. To obtain a differentiable objective, one can either marginalize over discrete variables directly (as done by Kingma et al. (2014) and in the STAN probabilistic programming system (Stan Development Team, 2013)), which doesn't scale over numbers of variables, or use a REINFORCE-style estimator (Williams, 1992; Mnih & Gregor, 2014), which tends to have high variance. A third approach, related to Bengio et al. (2013), is to represent discrete latent variables defined on a finite domain using a one-hot encoding, then relaxing them to a continuous probability simplex when used as an input to a recognition network. For example, when $\mathbf{y}$ is a one-hot encoding of a discrete value used in a recognition network which factors as $q_\phi(\mathbf{y} \mid \mathbf{x})q_\phi(\mathbf{z} \mid \mathbf{y}, \mathbf{x})$, then $q_\phi(\mathbf{y} \mid \mathbf{x})$ is itself a discrete distribution with a probability vector $\rho = g_\phi(\mathbf{x})$ for some deterministic function $g_\phi$. The value $\mathbf{y}$ is itself an input to a second function $h_\phi(\mathbf{x}, \mathbf{y})$ producing the parameters for $q_\phi(\mathbf{z} \mid \mathbf{y}, \mathbf{x})$. Instead of evaluating $h_\phi(\mathbf{x}, \mathbf{y})$ at a sampled value $\mathbf{y}$ (or enumerating over the entire domain), we simply evaluate it at the single point $\rho$, noting that $\rho = \mathbb{E}_{q_\phi(\mathbf{y}|\mathbf{x})}[\mathbf{y}]$. This may seem a crude approximation, replacing integration with a single evaluation, claiming $\mathbb{E}_{q_\phi(\mathbf{y}|\mathbf{x})}[h_\phi(\mathbf{x}, \mathbf{y})] \approx h_\phi(\mathbf{x}, \mathbb{E}_{q_\phi(\mathbf{y}|\mathbf{x})}[\mathbf{y}])$, which is not true in general for $h_\phi(\cdot)$. However, if $\rho$ is actually a one-hot encoding, i.e., when $\mathbb{E}_{q_\phi(\mathbf{y}|\mathbf{x})}[\mathbf{y}]$ has a single non-zero value, they are in fact equal. For our experiments we employ this plug-in estimator where applicable, although our framwork can express any of the above methods.

## 4 EXPERIMENTS

We evaluate our framework on along a number of different axes, pertaining to its ability to (i) learn *disentangled* representation from a little supervision, (ii) demonstrate capability at a relevant classification/regression task, (iii) successfully also learn the generative model, and (iv) admit the use of latent spaces of varying dimensionality Note that we do not set out to build the best possible classifier in these tasks. Instead, the classification task is a means to the end of demonstrating that the learned representation is indeed disentangled, often with minimal supervision. Also, details of neural network architectures, graphical models for the recognition networks, dataset characteristics, and hyper-parameter settings are provided in the Appendix.

### 4.1 MNIST AND SVHN

To begin with, we explore the facets of our model in the standard MNIST and Google Street-View House Numbers (SVHN) datasets. We use this example to highlight how the provision of even the slightest structure, coupled with minimal supervision, in often sufficient to induce the emergence of disentangled representations in the recognition network. Figure 3 shows the structure of the generative and recognition models for this experiment.

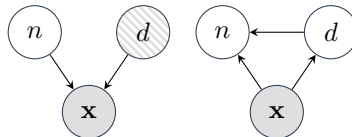

Figure 3: (left) Generative and (right) recognition model with digit $d$ and style $n$.

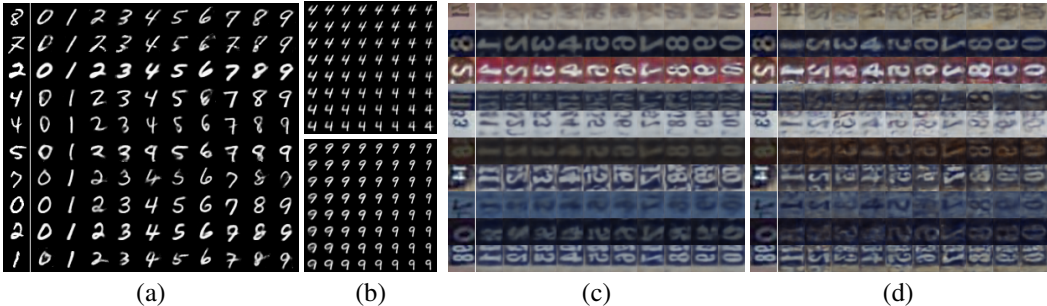

|     (a)      |     (b)      |     (c)      |     (d)      |

Figure 4: (a) Visual analogies for the MNIST data, with inferred style latent variable fixed and the label varied. (b) Exploration in "style" space for a 2D latent gaussian random variable. Visual analogies for the SVHN data when (c) fully supervised, and (d) supervised with just 100 labels/digit.

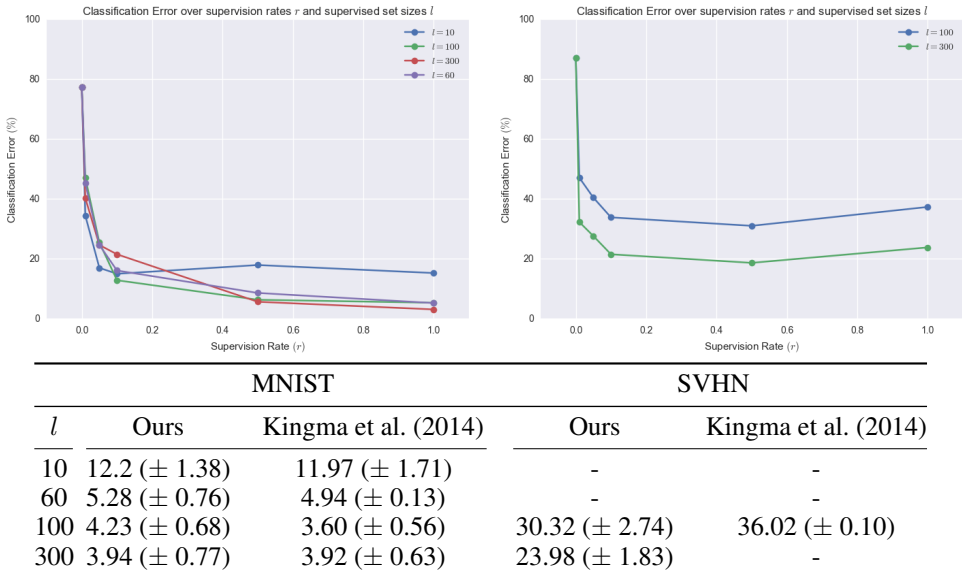

| | MNIST | | SVHN | |
| $l$ | Ours | Kingma et al. (2014) | Ours | Kingma et al. (2014) |
|---|---|---|---|---|
| 10 | 12.2 ($\pm$ 1.38) | 11.97 ($\pm$ 1.71) | - | - |
| 60 | 5.28 ($\pm$ 0.76) | 4.94 ($\pm$ 0.13) | - | - |
| 100 | 4.23 ($\pm$ 0.68) | 3.60 ($\pm$ 0.56) | 30.32 ($\pm$ 2.74) | 36.02 ($\pm$ 0.10) |
| 300 | 3.94 ($\pm$ 0.77) | 3.92 ($\pm$ 0.63) | 23.98 ($\pm$ 1.83) | - |

Figure 5: (Top) Classification error graphs over different labelled set (per class) sizes and supervision rates for MNIST (left) and SVHN (right). Note the steep drop in error rate with just a handful of labels per class ($l$), seen just a few times ($r$). (Bottom) Classification error rates for different (per-class) labelled-set sizes $l$ over different runs.

Figure 4(a) and (c) show the effect of first transforming a given input (leftmost column) into the disentangled latent space, and with the style latent variable fixed, manipulating the digit through the generative model to produce appropriately modified reconstructions. These were both derived with full supervision over a 50 and 100 dimensional Gaussian latent space for the styles, respectively. Figure 4(b) shows the transformation for a fixed digit, when the style latent is varied. This was derived with a simple 2D Gaussian latent space for the style. The last part, Figure 4(d) shows the ability of the network to begin disentangling the latent space with just 100 labelled samples per digit (training dataset size is 73000 points). Separation between style and class is clearly evident even with such little supervision.

We compute the classification accuracy of the label-prediction task with this model for both datasets, and the results are reported in the bottom of Figure 5. The results are compared to those reported in Kingma et al. (2014). For the MNIST dataset, we compare against model M2 as we run directly on the data, without performing a preliminary feature-extraction step. For the SVHN dataset, we compare against model M1+M2 even though we run directly on the data, using a CNN to simultaneously learn to extract features. Confidence estimates for both were computed off of 10 runs. We note that we fare comparably with these models, and in particular, when employing a CNN for feature extraction for the SVHN dataset, comfortably exceed them.

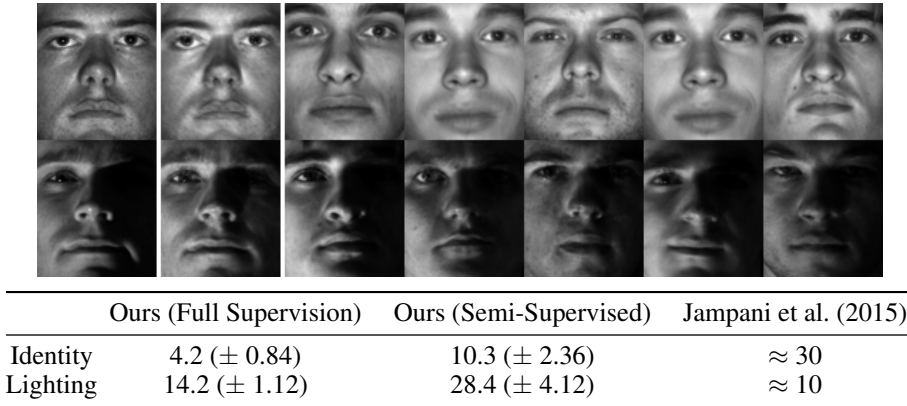

|  | Ours (Full Supervision) | Ours (Semi-Supervised) | Jampani et al. (2015) |
|---|---|---|---|
| Identity | 4.2 ($\pm$ 0.84) | 10.3 ($\pm$ 2.36) | $\approx 30$ |
| Lighting | 14.2 ($\pm$ 1.12) | 28.4 ($\pm$ 4.12) | $\approx 10$ |

Figure 7: (Top) Exploring the generative capacity of the model. Column 1: input image. Column 2: reconstruction. Columns 3-7: reconstructions with fixed (inferred) lighting and varying identities. (Bottom) Classification and regression error rates for the identity and lighting latent variables, fully-supervised, and semi-supervised with 20 distinct labelled example per variation axis (60 total). Classification is a direct 1-out-of-38 choice, whereas for the comparison, error is a nearest-neighbour loss based on the inferred reflectance. Regression loss for lighting is measured as cosine angle distance. Results for Jampani et al. (2015) are estimated from plot asymptotes.

Figure 5 shows the effect of the supervision rate $r$ on the error rate. As evident from the graph, the rate has a strong affect on how quickly one learns an effective classifier. This indicates that when labels are sparse or hard to come by, a training regime that runs largely unsupervised, even only occasionally looking at the supervised data, still learns to disentangle the latent-space representations.

## 4.2 INTRINSIC FACES

We next move to a harder problem involving a generative model of faces, attempting to highlight how the introduction of stronger dependency structures in the recognition model helps disentangle latents, particularly when the generative model assumes conditional independence between the latents. Here, we use the "Yale B" dataset as processed by Jampani et al. (2015) to train the models shown in Figure 6. The primary tasks we are interested in here are (i) the ability to manipulate the inferred latents to evaluate if they qualitatively achieve semantically meaningful disentangled representations, (ii) the classification of person identity, and (iii) the regression for lighting direction.

Figure 7 presents both qualitative and quantitative evaluation of the framework to jointly learn both the structured recognition model, and the generative model parameters. A particular point of note is that we explicitly encode "identity" as a categorical random variable since we have knowledge about the domain and the relevant axis to explore. Since we also learn the generative model, which in the domain of the actual dataset is simply the expression $(n.l) \times r + \epsilon$, we can afford to weakly specify the structure allowing for some neural-network component to take up the requisite slack in order to reconstruct the input. This allows us to directly address the task of predicting identity, instead of approaching it through surrogate evaluation methods (e.g. nearest-neighbour classification based on inferred reflectance).

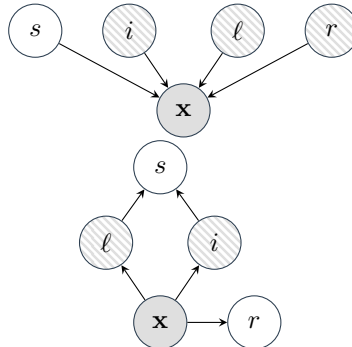

Figure 6: (Top) Generative and (Bottom) recognition model with identity $i$, lighting $l$, reflectance $r$, and shading $s$.

While this formulation allows us to to perform the identity classification task, the fact that our recognition model never supervises the reflectance means that the variable can typically absorb some of the representational power of other, semi-supervised nodes. This is particularly the case when dealing with high-dimensional latent spaces as for reflectance and shading.

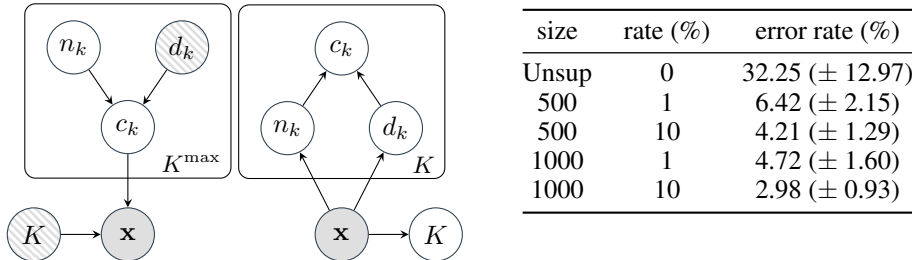

| size | rate (%) | error rate (%) |
|---|---|---|
| Unsup | 0 | 32.25 ($\pm$ 12.97) |
| 500 | 1 | 6.42 ($\pm$ 2.15) |
| 500 | 10 | 4.21 ($\pm$ 1.29) |
| 1000 | 1 | 4.72 ($\pm$ 1.60) |
| 1000 | 10 | 2.98 ($\pm$ 0.93) |

Figure 8: Generative (l) and recognition (m) model with digit $d$, style $n$, canvas $c$, and count $K$.

### 4.3 MULTI-MNIST

Finally, we run an experiment to test the ability of our framework to handle models that induce latent representations of *variable* dimension. We extend the simple model from the MNIST experiment by composing it with a *stochastic* sequence generator, to test its ability to count the number of digits in a given input image, given its ability to encode and reconstruct the digits in isolation. The graphical models employed are depicted in Figure 8.

We observe that we are indeed able to reliable learn to count, at least within the limits of upto 3 digits in the multi-mnist dataset. The dataset was generated directly from the MNIST dataset by manipulating the scale and positioning of the standard digits into a combined canvas, evenly balanced across the counts and digits. The results across different supervised set sizes and supervision rates are shown in the table in Figure 8.

## 5 DISCUSSION AND CONCLUSION

In this paper, we introduce a general framework for semi-supervised learning in the VAE setting that allows incorporation of graphical models to specify a wide variety of structural constraints on the recognition network. We demonstrate its flexibility by applying it to a variety of different tasks in the visual domain, and evaluate its efficacy at learning disentangled representations in a semi-supervised manner, showing strong performance.

This framework ensures that the recognition network learns to make predictions in an interpretable and disentangled space, constrained by the structure provided by the graphical model. The structured form of the recognition network also is typically a better fit for vision models, as it helps better capture complexities in the likelihood (usually the renderer). Given that we encode graphical models in the recognition network, and Johnson et al. (2016) encode it in the generative model in concert with VAEs, a natural extension would be the exploration of the ability to learn effectively when specifying structure in *both* by means of graphical models. This is a direction of future work we are interested in, particularly in context of semi-supervised learning.

The framework is implemented as a Torch library (Collobert et al., 2011), enabling the construction of stochastic computation graphs which encode the requisite structure and computation. This provides another direction to explore in the future – the extension of the stochastic computation graph framework to probabilistic programming (Goodman et al., 2008; Wingate et al., 2011; Wood et al., 2014). Probabilistic programs go beyond the presented framework to include stochastic inference and the ability to specify arbitrary models of computation. The combination of such frameworks with neural networks has recently been studied in Ritchie et al. (2016); Le et al. (2016), and indicates a promising avenue for further exploration.

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

## APPENDIX

### FORMULATION

**Gradients of the Variational Objective:**  We consider the gradients of the form in Equation (4) with respect to $\theta, \phi_{\mathbf{z}}$, and $\phi_{\mathbf{y}}$. In particular, note that for both $\theta$ and $\phi_{\mathbf{z}}$ the gradient is the same as the gradient with respect to the conditional ELBO $\mathcal{L}_{\mathbf{x}|\mathbf{y}}$, up to a per-datapoint scaling factor $q(\mathbf{y}^i \mid \mathbf{x}^i)$. For continuous latent variables, as well as for many discrete random variables, the expectations over $\mathbf{z}$ can be reparameterized into a form where the gradients can be approximated with a single sampled value. Evaluating Equation (4) at this point yields estimators for the ELBO $\widehat{\mathcal{L}}$ and conditional ELBO $\widehat{\mathcal{L}}_{\mathbf{x}|\mathbf{y}}$, as well as corresponding single-sample gradient estimates $\widehat{\nabla}\mathcal{L}$ and $\widehat{\nabla}\mathcal{L}_{\mathbf{x}|\mathbf{y}}$ for each set of parameters.

Gradient estimates for $\theta$ and $\phi_{\mathbf{z}}$ are proportional to the gradients of the conditional ELBO as

$$\widehat{\nabla}_\theta\, \mathcal{L}(\theta, \phi; \mathbf{x}^i)\big|_{\mathbf{y}=\mathbf{y}^i} = q_{\phi_{\mathbf{y}}}\big(\mathbf{y}^i \mid \mathbf{x}^i\big)\widehat{\nabla}_\theta\, \mathcal{L}_{\mathbf{x}|\mathbf{y}},$$
$$\widehat{\nabla}_{\phi_{\mathbf{z}}}\, \mathcal{L}(\theta, \phi; \mathbf{x}^i)\big|_{\mathbf{y}=\mathbf{y}^i} = q_{\phi_{\mathbf{y}}}\big(\mathbf{y}^i \mid \mathbf{x}^i\big)\widehat{\nabla}_{\phi_{\mathbf{z}}}\, \mathcal{L}_{\mathbf{x}|\mathbf{y}},$$

while the gradient with respect to the "classifier" parameters $\phi_{\mathbf{y}}$ takes a different form. Applying the product rule to Equation (4) we have

$$\widehat{\nabla}_{\phi_{\mathbf{y}}}\, \mathcal{L}(\theta, \phi; \mathbf{x}^i)\big|_{\mathbf{y}=\mathbf{y}^i}$$
$$= \Big[\widehat{\mathcal{L}}_{\mathbf{x}|\mathbf{y}} + \log p\big(\mathbf{y}^i\big) - \log q_{\phi_{\mathbf{y}}}\big(\mathbf{y}^i \mid \mathbf{x}^i\big)\Big]\nabla_{\phi_{\mathbf{y}}} q_{\phi_{\mathbf{y}}}\big(\mathbf{y}^i \mid \mathbf{x}^i\big) - q_{\phi_{\mathbf{y}}}\big(\mathbf{y}^i \mid \mathbf{x}^i\big)\nabla_{\phi_{\mathbf{y}}} \log q_{\phi_{\mathbf{y}}}\big(\mathbf{y}^i \mid \mathbf{x}^i\big)$$
$$= \Big[\widehat{\mathcal{L}}_{\mathbf{x}|\mathbf{y}} + \log p\big(\mathbf{y}^i\big) - \log q_{\phi_{\mathbf{y}}}\big(\mathbf{y}^i \mid \mathbf{x}^i\big) - 1\Big]\nabla_{\phi_{\mathbf{y}}} q_{\phi_{\mathbf{y}}}\big(\mathbf{y}^i \mid \mathbf{x}^i\big)$$
$$= q_{\phi_{\mathbf{y}}}\big(\mathbf{y}^i \mid \mathbf{x}^i\big)\Big[\widehat{\mathcal{L}} - 1\Big]\nabla_{\phi_{\mathbf{y}}} \log q_{\phi_{\mathbf{y}}}\big(\mathbf{y}^i \mid \mathbf{x}^i\big).$$

## MODEL AND NETWORK PARAMETERS

We note for that all the experiments, save the one involving Street-View House Numbers (SVHN), were run using a 2-3 layer MLP with 512 nodes and using a Bernoulli loss function. For SVHN, we additionally employed a two stage convolutional and a 2 stage deconvolutional network to effectively extract features for the standard MLP model for the recognition network and the generative model respectively; training the entire network end-to-end. For learning, we used AdaM (Kingma & Ba, 2014) with a learning rate of 0.001 (0.0003 for SVHN) and momentum-correction terms set to their default values. As for the minibatch sizes, they varied from 80-500 depending on the dataset being used and its size.

## MODELS

The syntax of our computation graph construction is such that the first call instantiates the computation, and the second instantiates the node and its connections. For specified random variables, the first set of parameters defines the prior and second set the parameters for the proposal distributions. In all our models, we extract the common, feature-extraction portions of the recognition model $q_\phi$ into a simple pre-encoder. Parameters and structure for this are specified above.

The class-conditional model for MNIST and SVHN.

```lua
local ndim = 50

local program = {}

function program:getNetwork()
  local input = nn.Identity()() -- required to make nngraph play nice
  -- the actual program
  local d = pp.DiscreteR({torch.Tensor(1,10):fill(1/10)})({input})
  local mu = nn.Sequential()
    :add(nn.JoinTable(2))
    :add(nn.FluidLinear(ndim))
    :add(nn.SoftPlus())
  local sig = nn.Sequential()
    :add(nn.JoinTable(2))
    :add(nn.FluidLinear(ndim))
    :add(nn.SoftPlus())
  local n = pp.Gaussian({
      torch.zeros(1,ndim),
      torch.zeros(1,ndim)
  })({mu({d, input}), sig({d, input})})
  -- end program
  nngraph.annotateNodes()       -- necessary to annotate nodes with local varnames
  return pp.gModule({input}, {d, n})
end

return program
```

The model used for the faces dataset.

```lua
local program = {}

function program:getNetwork()
  local input = nn.Identity()() -- required to make nngraph play nice
  -- the actual program
  local id = pp.DiscreteR({torch.Tensor(1,38):fill(1/38)})({input})
  local light = pp.Gaussian({
      torch.zeros(1,3),
      torch.zeros(1,3)
  })({pp.r(input), pp.r(input)})
  local factorQ = nn.Sequential()
    :add(nn.JoinTable(2))
    :add(nn.FluidLinear(20))
    :add(nn.SoftPlus())
  local shading = pp.Gaussian({
      torch.zeros(1,20),
      torch.zeros(1,20)
  })({pp.r(factorQ({id,light})), pp.r(factorQ({id,light}))})
  local reflectance = pp.Gaussian({
      torch.zeros(1,20),
      torch.zeros(1,20)
  })({pp.r(input), pp.r(input)})
  -- end program
  nngraph.annotateNodes()       -- necessary to annotate nodes with local varnames
  return pp.gModule({input}, {shading, reflectance})
end

return program
```

The model used for the multi-mnist dataset.

```lua
local program = {}

local function mnist()
  local input = nn.Identity()() -- required to make nngraph play nice
  local d = pp.DiscreteR({torch.Tensor(1,10):fill(0.1)})({input})
  local n = pp.Gaussian({
      torch.zeros(1,50),
      torch.zeros(1,50)
  })({pp.r(input), pp.r(input)})
  -- end program
  nngraph.annotateNodes()        -- necessary to annotate nodes with local varnames
  return pp.gModule({input}, {d, n})
end

function program:getNetwork()
  local input = nn.Identity()() -- required to make nngraph play nice
  -- the actual program
  local c = pp.Discrete(({torch.Tensor(1,5):fill(0.2)})({input}))
  -- needswork: have to handle number of inputs and inter-repeat-state
  local ds = pp.Repeat(mnist())({input, c})
  -- end program
  nngraph.annotateNodes()        -- necessary to annotate nodes with local varnames
  return pp.gModule({input}, {ds})
end

return program
```

