# Peer review of "Learning Disentangled Representations in Deep Generative Models"

_ICLR 2017 — rejected_

[Official Review · AnonReviewer2 · rating 6 · confidence 5 · 17 Dec 2016]

This paper investigates deep generative models with multiple stochastic nodes and gives them meaning by semi-supervision. From a methodological point of view, there is nothing fundamentally novel (it is very similar to the semi-supervised work of Kingma et al; although this work has sometimes more than two latent nodes, it is not a complex extension). There is a fairly classical auxiliary variable trick used to make sure the inference network for y is trained over all data points (by supposing y is in fact is a latent variable with an observation \tilde y; the observation is y if y is observed, or uninformative for unobserved y). Alternatively, one can separate the inference used to learn the generative model (which throws out inference over y if it is observed), from an inference used to 'exercise' the model (approximate the complex p(y|x) in the model by a simpler q(y|x) - effectively inferring the target p(y|x) for the data where only x is collected). Results are strong, although on simple datasets. Overall this is a well written, interesting paper, but lacking in terms of methodological advances.

Minor:
- I feel the title is a bit too general for the content of the paper. I personally don't agree with the strong contrast made between deep generative models and graphical models (deep generative models are graphical models, but they are more typically learned and un-interpretable than classical graphical models; and having multiple stochastic variables is not exclusive to graphical models, see DRAW, Deep Kalman Filter, Recurrent VAE, etc.). The word 'structure' is a bit problematic; here, the paper seems more concerned with disentangling and semanticizing the latent representation of a generative model by supervision. It is debatable whether the models themselves have structure.

[Official Review · AnonReviewer3 · rating 6 · confidence 4 · 17 Dec 2016]
**A variant of the semi-supervised VAE.**

This paper proposed a variant of the semi-supervised VAE model which leads to a unified objective for supervised and unsupervised VAE.  This variant gives software implementation of these VAE models more flexibility in specifying which variables are supervised and which are not.

This development introduces a few extra terms compared to the original semi-supervised VAE formulation proposed by Kingma et al., 2014.  From the experiment results it seems that these terms do not do much as the new formulation and the performance difference between the proposed method and Kingma et al. 2014 are not very significant (Figure 5).  Therefore the benefit of the new formulation is likely to be just software engineering flexibility and convenience.

This flexibility and convenience is nice to have, but it is better to demonstrate a few situations where the proposed method can be applied while for other previous methods it is non-trivial to do.

The paper's title and the way it is written make me expect a lot more than what is currently in the paper.  I was expecting to see, for example, structured hidden variable model for the posterior (page 4, top), or really "structured interpretation" of the generative model (title), but I didn't see any of these.  The main contribution of this paper (a variant of the semi-supervised VAE model) is quite far from these.

Aside from these, the plug-in estimation for discrete variables only works when the function h(x,y) is a continuous function of y.  If however, h(x, y) is not continuous in y, for example h takes one form when y=1 and another form when y=2, then the approach of using Expectation[y] to replace y will not work.  Therefore the "plug-in" estimation has its limitations.

[Official Review · AnonReviewer1 · rating 5 · confidence 3 · 17 Dec 2016]
**No Title**

This paper introduces a variant of the semi-supervised variational auto-encoder (VAE) framework. The authors present a way of introducing structure (observed variables) inside the recognition network.

I find that the presentation of the inference with auxiliary variables could be avoided, as it actually makes the presentation unnecessarily complicated. Specifically, the expressions with auxiliary variables are helpful for devising a unified implementation, but modeling-wise one can get the same model without these auxiliary variables and recover a minimal extension of VAE where part of the generating space is actually observed. The observed variables mean that the posterior needs to also condition on those, so as to incorporate the information they convey. The way this is done in this paper is actually not very different from Kingma et al. 2014, and I am surprised that the experiments show a large deviation in these two methods' results. Given the similarity of the models, it'd be useful if the authors could give a possible explanation on the superiority of their method compared to Kingma et al. 2014. By the way, I was wondering if the experimental setup is the same as in Kingma et al. 2014 for the results of Fig. 5 (bottom) - the authors mention that they use CNNs for feature extraction but from the paper it's not clear if Kingma et al. do the same. 

On a related note, I was wondering the same for the comparison with Jampani et al. 2015. In particular, is that model also using the same rate of supervision for a fair comparison?

The experiment in section 4.3 is interesting and demonstrates a useful property of the approach.

The discussion of the supervision rate (and the pre-review answer) is helpful in giving some insight about what is a successful training protocol to use in semi-supervised learning.

Overall, the paper is interesting but the title and introduction made me expect something more from it. From the title I expected a method for interpreting general deep generative models, instead the described approach was about a semi-supervised variant of VAE - naturally including labelled examples disentangles the latent space, but this is a general property of any semi-supervised probabilistic model and not unique to the approach described here. Moreover, from the intro I expected to see a more general approximation scheme for the variational posterior (similar to Ranganath et al. 2015  which trully allows very flexible distributions), however this is not the case here.

Given the above, the contributions of this paper are in defining a slight variant of the semi-supervised VAE, and (perhaps more importantly) formulating it in a way that is amendable to easier automation in terms of software. But methodologically there is not much contribution to the current literature. The authors mention that they plan to extend the framework in the probabilistic programming setting. It seems indeed that this would be a very promising and useful extension. 

Minor note: three of Kingma's papers are all cited in the main text as Kingma et al. 2014, causing confusion. I suggest using Kingma et al. 2014a etc.

[Public Comment · Siddharth N · 12 Jan 2017]
**Response to reviewers**

We thank the reviewers for their comments and suggestions and are encouraged by
the positive feedback regarding its value, interest, relevance, and clarity.

In this top-level response, we address the two central issues raised by the
reviewers.

- The title and the semantics of 'structure'

  The reviewers found the title somewhat confusing and perhaps overly general in
  addition to potential confusion due the many meanings of the term 'structure'.
  To ameliorate this issue, we have done the following:

  a. Title change

     We changed the title to
     	``Learning Disentangled Representations in Deep Generative Models''
     to better reflect the contributions of the submission.

  b. Clarification for 'structure'

     We realise that the term 'structure' can have multiple meanings, even
     within the confines of graphical and generative models.

     Our use of the term is intended to refer to the (arbitrary) dependencies
     one would like to employ in the recognition model, particularly in regard
     to there being consistent 'interpretable' semantics of what the variables
     in the model represent.

     We have updated the abstract and the introduction (Section 1, end) in the
     manuscript making this clarification and removing extraneous instances.

- Relation to Kingma.et.al 2014 [1]

  We would like to note here that we do not simply reformulate the
  semi-supervised work in [1]. Our formulation is not a trivial extension of
  [1], rather, it allows us to extend semi-supervised learning to a broader
  class of latent-variable models.
  We list below the important distinctions:

  a. Continuous-domain semi-supervision

     We can handle partial labels for continuous random variables, not just
     discrete ones. In this case, the factorisation in Eqn 4 corresponds to a
     *regressor* instead of a classifier. The work in [1] requires
     marginalization over the partially-observed variable’s support in the
     unsupervised case, which means that latent variables must in practice be
     discrete.

     Indeed we make use of continuous latent variables in the Faces experiment
     (Section 4.2), where the lighting is a partially supervised 3-dimensional
     Gaussian random variable.

  b. Scaling the classifier term

     The formulation in Eqn 4 naturally incorporates the classifier term, as
     opposed to a separate fixed hyper-parameter (alpha, in Eqn 9 in [1]) that
     controls the contribution of the classifier term in the objective. 

     This makes the formulation more flexible and general purpose for different
     factorisations of the variational approximations used.

  c. Framework for  implementation of models

     As pointed out by the reviewers, our formulation allows for easy automated
     implementation of a wide variety of models. This is in the same spirit as a
     number of approaches such as Automatic Differentiation (AD) and
     Probabilistic Program inference, where the choice of a particular means of
     representation enables ease of automation for a great variety of different
     cases.

     The ease with which one can describe even minimally complex models, such as
     that employed for the Multi-MNIST experiments, is shown in the Appendix.

  d. Semi-supervised learning on varying subsets of variables

     A particular benefit of the automation mentioned above, is that we derive
     the ability to partially supervise any subset of variables, regardless of
     type. Indeed we even have the ability to supervise *different* latent
     variables for different data points by virtue of our formulation
     automatically factorising into labelled and unlabelled terms, on a
     per-data-point (or minibatch) basis. This is a particularly desirable
     characteristic to have in dealing with missing-data issues often
     encountered in large datasets.

We have updated the Framework and Formulation section (Section 3, after Eqn 4)
in the manuscript with a description of these differences.

We address the remainder of the comments by the reviewers in the corresponding
responses to the reviews.

[1] Diederik P Kingma, Shakir Mohamed, Danilo Jimenez Rezende, and Max Welling.
    Semi-supervised learning with deep generative models. In Advances in Neural
    Information Processing Systems, pp. 3581–3589, 2014

[Final Decision · Program Chairs · 06 Feb 2017]
**ICLR committee final decision**

The paper is a clearly presented application of deep generative models in the semi-supervised setting. After reviewing the discussion and responses, the reviewers felt that the paper while interesting, is limited in scope, and unfortunately not yet ready for inclusion in this year's proceeding.